# Influence of Bioinspired Lithium-Doped Titanium Implants on Gingival Fibroblast Bioactivity and Biofilm Adhesion

**DOI:** 10.3390/nano11112799

**Published:** 2021-10-22

**Authors:** Aya Q. Alali, Abdalla Abdal-hay, Karan Gulati, Sašo Ivanovski, Benjamin P. J. Fournier, Ryan S. B. Lee

**Affiliations:** 1School of Dentistry, The University of Queensland, Herston, QLD 4006, Australia; a.alali@uqconnect.edu.au (A.Q.A.); abdalla.ali@uq.edu.au (A.A.-h.); k.gulati@uq.edu.au (K.G.); s.ivanovski@uq.edu.au (S.I.); 2Department of Engineering Materials and Mechanical Design, Faculty of Engineering, South Valley University, Qena 83523, Egypt; 3Laboratory of Molecular Oral Physiopathology, INSERM UMRS 1138, Cordeliers Research Center, 75006 Paris, France

**Keywords:** titanium, implants, nanostructure, gingival fibroblasts, biofilm, soft-tissue integration, surface modification

## Abstract

Soft tissue integration (STI) at the transmucosal level around dental implants is crucial for the long-term success of dental implants. Surface modification of titanium dental implants could be an effective way to enhance peri-implant STI. The present study aimed to investigate the effect of bioinspired lithium (Li)-doped Ti surface on the behaviour of human gingival fibroblasts (HGFs) and oral biofilm *in vitro*. HGFs were cultured on various Ti surfaces—Li-doped Ti (Li_Ti), NaOH_Ti and micro-rough Ti (Control_Ti)—and were evaluated for viability, adhesion, extracellular matrix protein expression and cytokine secretion. Furthermore, single species bacteria (*Staphylococcus aureus*) and multi-species oral biofilms from saliva were cultured on each surface and assessed for viability and metabolic activity. The results show that both Li_Ti and NaOH_Ti significantly increased the proliferation of HGFs compared to the control. Fibroblast growth factor-2 (FGF-2) mRNA levels were significantly increased on Li_Ti and NaOH_Ti at day 7. Moreover, Li_Ti upregulated COL-I and fibronectin gene expression compared to the NaOH_Ti. A significant decrease in bacterial metabolic activity was detected for both the Li_Ti and NaOH_Ti surfaces. Together, these results suggest that bioinspired Li-doped Ti promotes HGF bioactivity while suppressing bacterial adhesion and growth. This is of clinical importance regarding STI improvement during the maintenance phase of the dental implant treatment.

## 1. Introduction

Peri-implant diseases of endo-osseous oral implants are mainly initiated by biofilm accumulation and subsequent host immuno-inflammatory responses at the transmucosal (implant abutment-mucosa) interface [1,2]. Peri-implant soft tissues exist as a physical barrier between the oral environment and the implant. However, the peri-implant mucosal seal could be considered fragile in disease, as it lacks the complex supra-crestal connective tissue structures usually found in the natural dentition [3], which may be responsible for the rapid rate of disease progression [4,5,6]. Hence, various attempts have been made to alter the surface of dental implants and abutments to augment soft tissue integration (STI), as reviewed elsewhere [7].

Surface characteristics of the implant, such as topography and chemistry, play a key role in determining tissue responses [8]. Most of the proposed Ti surface topographical modifications aim to improve osseointegration quantity and quality by altering the surface roughness at microscale levels [9,10,11,12], while others focus on creating implant devices with antimicrobial properties by mimicking self-cleansing surfaces found in nature [13,14,15]. Furthermore, effective bactericidal functions have been achieved by incorporating nano-structures onto implant substrates, such as nanopillars [16,17] or local antibiotic releasing nanotubes fabricated via anodisation [18,19]. With regard to the peri-implant soft tissue’s response, previous *in vitro* studies using gingival human fibroblasts demonstrated an increase in the proliferation [20], mechanical stimulation [21], collagen production [22] and attachment [23] to the substrates, indicating the potential for nanostructures to promote connective tissue formation around an implant.

Nature-inspired nano-topographies are commonly reported in the literature, and reproduction of the shape and arrangement of natural nanoscale patterns have been attempted to improve the characteristics of biomaterials, notably in implants research [24,25,26]. The extracellular matrix (ECM) of biological tissues has been a source of inspiration for several studies [24,26], mainly targeting the arrangement of laminin and collagen nano-fibres [27]. Scaffolds with ECM-like features exhibit an increase in the cellular deposition of hydroxyapatite, which aids in promoting the mineralisation required for osseointegration [28]. Moreover, these features influence initial filopodia-surface interactions [29].

Studies have revealed that chemical surface treatments [30], such as hydrothermal alkalinisation [31], electrochemical anodisation [23], and electrochemical oxidization [32], are cost-effective strategies to nano-engineer surfaces on Ti-based implants [31]. Chemically induced nanostructures have been shown to increase gingival fibroblast attachment while inhibiting bacterial adhesion *in vitro* [33]. The incorporation of metal ions, such as zinc [34], magnesium [35] and Li [36], into Ti surfaces enhances various cellular activities in osteoblasts and fibroblasts [37,38]. Our group recently reported the utilisation of hydrothermal transformation to fabricate Li-doped Ti with sustainable Li+ ions release [39]. Lithium belongs to the alkali metal group and is considered a biologically functional ion. It has been shown that Li ions can stimulate bone growth and periodontal ligament cell differentiation through the *Wnt*/β-catenin signalling pathway [40,41]. Abdal-hay et al. [39] investigated the influence of different LiCl concentrations on surface properties of doped Li-Ti. Their results show that a Li-Ti porous layer with nanostructure characteristics was nucleated and formed on the Ti surface. Furthermore, Li-incorporated Ti exhibits improved wettability and mechanical stability compared to untreated Ti surfaces, with an improved effect on osteoblast activity [36,39]. The impacts of Li-incorporated surface modification on gingival fibroblasts, however, have yet to be extensively explored.

An ideal implant surface should modulate cellular responses, leading to the timely establishment and maintenance of osseointegration, soft-tissue integration and prevention of bacterial adhesion. The current study explores the STI and antibacterial functions of ECM-mimicking nanoscale Li-Ti surfaces as the next generation of modified Ti dental implants.

## 2. Materials and Methods

### 2.1. Titanium Surface Modification

A 99.5% Ti flat foil (0.3 mm thickness) was purchased from Nilaco Corporation (Tokyo, Japan). Ti foil was mechanically treated using a gradient of sandpapers to form a micro-machining-like surface topography (Control_Ti) [42]. Ti foil was cut into 10 mm × 10 mm squares using diamond EXAKT’s saw machine. Next, Ti was etched in an acid mixture (equal volumes of concentrated acids and water H_2_SO_4_: HCl: H_2_O) at 80 °C for 1 h to remove the natural oxide layer and increase surface roughness, followed by immersion in 200 mL of 5.0 M NaOH aqueous solution at 60 °C for 24 h, and then rinsed with distilled water. To introduce Li ions, the alkali-treated Ti samples were first immersed in lithium chloride (LiCl: 0.025 M), then hydrothermally treated in a Teflon container at 90 °C for 24 h. After Li-containing compound precipitation, the Ti substrates were rinsed in distilled water and dried at 45 °C for 24 h [39]. The substrates were then grouped according to the treatment: (1) lithium-incorporated alkaline-treated Ti (Li_Ti) as a test group, (2) alkaline-treated Ti (NaOH_Ti) as a test group, and (3) mechanically prepared micro-rough Ti (Control_Ti). All surfaces to be tested were sterilised by immersion in 70% ethanol for 8 h followed by air drying for 24 h, and ultraviolet irradiation for 30 min each side. To observe the topography, titanium substrates were mounted on a holder with double-sided conductive tape, coated with 10 nm platinum, and at least 5 substrates of each group were viewed under SEM (SEM, JSM- 7001F, Joel, Tokyo, Japan).

### 2.2. Culture of Human Gingival Fibroblasts

Primary human gingival fibroblast cells cultured at passages 4–6 were used for all experiments. All subjects gave their informed consent for inclusion before they participated in the study. The study was conducted in accordance with the Declaration of Helsinki, and the protocol was approved by the University of Queensland Institutional Human Ethics Research Committee (No. 2019000134). The cells were cultured at 37 °C and 5% CO_2_ in Dulbecco’s modified Eagle’s medium (DMEM, Life Technologies, Scoresby, VIC, Australia) supplemented with 10% foetal bovine serum (FBS from Gibco^®^, Clayton, VIC, Australia) and 1% Penicillin-Streptomycin-Glutamine (Gibco^®^, Clayton, VIC, Australia). Cells were grown in Corning^®^ T75 Flasks (Thermo Fisher Scientific, Madrid, Spain), and upon 80% confluency, detached using 0.04% trypsin, and then seeded at a density of 5000 cells per Ti substrate in 12-well plates. The LIVE/DEAD assay^®^ (Life Technologies, Scoresby, VIC, Australia) was performed according to the manufacturer’s instructions to assess cell viability. At predetermined timepoints, cultured samples were washed twice with phosphate-buffered saline (PBS), then incubated with fluorescein diacetate (FDA/live; 1:200) and propidium iodide (PI/dead) diluted in PBS, for 20 min at 37 °C and 5% CO_2_. Images of the stained cultures were obtained using confocal microscopy (Nikon Eclipse Ti-E. Nikon Instruments Inc., Melville, NY, USA).

### 2.3. Cell Attachment and Spread Morphology

After 4, 24 h and 7 days incubation, cells on the different Ti substrates were fixed for 20 min with 4% paraformaldehyde in PBS (PFA). After washing twice with PBS, cells were permeabilised with Triton X-100 (0.5%) in PBS for 10 min, followed by incubation in blocking buffer (10% Bovine Serum Albumin, Glycine, tween, and PBS) for 1 h. The primary antibody for collagen I (1:250) was then added for one hour at room temperature. After three PBS washes, secondary antibodies (Goat An-ti-Mouse/Rabbit Alexa fluor 488,568), DAPI Staining Solution (ab228549) (1:500), and Phalloidin-California Red Conjugate (1:1000) were added, and samples were incubated in the dark for 30 min. After a final three PBS washes, images of each sample were obtained using confocal microscopy (Nikon Eclipse Ti-E. Nikon Instruments Inc. USA), and image analysis was performed using Image J (Fiji V1.53 g, National Institutes of Health, Bethesda, MD, USA).

For surface morphology and spreading observation, cultured samples were fixed with 4% PFA for 20 min, washed twice in sodium cacodylate buffer and immersed in glutaraldehyde for 30 min, rinsed twice in sodium cacodylate buffer, dehydrated in multiple concentrations of ethanol (20–100%), then immersed in hexamethyldisilazane (HDMS) for 30 min. Finally, samples were left to fully dry before coating with 10 nm platinum for SEM imaging (JSM-7001F, Jeol, Tokyo, Japan).

### 2.4. Cell Count

Ti substrates were placed in 24 wells containing 350 μL of proteinase K (Invitrogen, Waltham, MA, USA; proteinase K/phosphate buffered EDTA (PBE) 0.5 mg/mL) for DNA content analysis, and were incubated overnight at 56 °C. Following this, 100 μL from each well was aliquoted in triplicate into a black 96-well plate, and 100 μL of the PicoGreen (P11496, Invitrogen, Waltham, MA, USA) working solution was added. Plates were incubated in the dark for 5 min before reading in a fluorescence plate reader (excitation 485 nm, emission 520 nm).

### 2.5. Gene Expression by Real-Time Quantitative Polymerase Chain Reaction (RT-qPCR)

Real-time qPCR was performed to determine changes in expression of selected genes by HGFs on the Ti samples. Briefly, RNA was extracted from HGFs (5 pooled samples, each sample 5000 cell/cm^2^) using TRIzol following the manufacturer’s instructions. Phase separation was performed to generate the aqueous phase, followed by RNA precipitates. cDNA synthesis was completed using a RevertAid First Strand cDNA Synthesis Kit (ThermoFisher Scientific, Scoresby, Australia). mRNA for collagen I, collagen III, CXCL8, IL_1β and FN was measured according to comparative CT values using the StepOnePlusTM Real-Time PCR system (Applied BiosystemsTM, ThermoFisher Scientific, Scoresby, Australia), and normalised against two reference genes, hGAPDH and h18 s. Forward and reverse primer sequences corresponding to each tested gene are listed in Table 1. Fold change analysis was standardised relative to control.

### 2.6. Extracellular Matrix Expression by Luminex

A Magnetic Luminex Screening Assay with a Human Premixed Multi-Analyte Kit (LXSAHM, R&D Systems Luminex^®^, Minneapolis, MN, USA) was utilised to assay conditioned media from human gingival fibroblast cultures for proteins of interest. The customised 5-plex panel included primary growth and inflammatory analytes: fibroblast growth factor (FGF)-2, matrix metalloproteinase (MMP)1, MMP8, vascular endothelial growth factor (VEGF)-A, and platelet-derived growth factor (PDGF)-BB, and Multiplex-ELISA was performed according to the manufacturer’s instructions. Triplicate supernatant samples were assayed in duplicate. Culture media were used as a negative control for all the samples.

### 2.7. Ethics Approval and Saliva Collection

Methicillin-sensitive *Staphylococcus aureus* (MSSA) was obtained from the American Type Culture Collection (ATCC; 25923, Manassas, VA, USA) for growing mono-species biofilms. Saliva from healthy volunteers was used for growing polymicrobial salivary biofilms. It was conducted in accordance with the Declaration of Helsinki, and the protocol was approved by the University of Queensland Institutional Human Ethics Research Committee (No. 2019001113). Informed consent was obtained from all subjects before sample collection. Unstimulated saliva from six healthy individuals was collected using a protocol previously reported [43]. In brief, volunteers were requested to provide approximately 2.0 mL of unstimulated saliva by spitting it into a 50 mL centrifuge tube. The volunteers had good gingival health, as evidenced by oral examination, and had not consumed antimicrobials and were not regularly using antimicrobial mouth rinses. The collected saliva was pooled, mixed with equal amounts of 70% glycerol stock solution and vortexed. The resultant mix was aliquoted into 1.5 mL Eppendorf tubes and stored at −80 °C until further processing.

### 2.8. Biofilm Development

#### 2.8.1. Single Species Biofilms

*S. aureus* was inoculated into 10 mL of brain heart infusion broth (BHI) in a centrifuge tube using Culti-Loops™, cultured overnight, then the tubes were centrifuged, and the supernatant was discarded. The sedimented bacteria were resuspended in sterile phosphate-buffered saline. The turbidity of the suspension was measured spectrophotometrically (Thermo Scientific™ GENESYS 10S UV-Vis spectrophotometer). The turbidity of the suspension was adjusted to approximately 1 × 10^7^ CFU/mL of *S. aureus*, which was subsequently used for culturing purposes.

#### 2.8.2. Multispecies Biofilms

Similarly, 1.0 mL of unstimulated saliva was mixed with 9 mL of BHI broth for overnight culturing. The inoculum was adjusted to 1 × 10^7^ CFU/mL as above.

#### 2.8.3. Biofilm Culture and Development

One millilitre of the bacteria was mixed with 8.0 mL BHI and 1.0 mL defibrinated sheep’s blood, and kept in an anaerobic gas box inside a shaker (80 rpm) at 37 Celsius overnight to allow bacterial growth. The following day, concentrations of bacteria were determined spectrophotometrically. Approximately 1 × 10^7^ CFU/mL of *S. aureus* or salivary biofilm were cultured separately over sterile Li_Ti, NaOH_Ti, and control substrates (*n* = 3) placed in a sterile 24-well tissue culture (Corning CLS3524, Thermo Fisher Scientific, Scoresby, Australia). At the predetermined time points, samples were washed twice with phosphate-buffered saline (pH 7.2) prior to further experiments.

### 2.9. Bacterial Metabolic Activity

An XTT (2,3-Bis-(2-methoxy-4-nitro-5-sulfophenyl)-2H-tetrazolium-5-carboxanilide) kit (Sigma-Aldrich, Castle-Hill, NSW, Australia) was used to test bacterial metabolic activity. In this process, 200 μg/mL of XTT was mixed with 25 μM of menadione. Ti substrates (*n* = 4) were washed with PBS, immersed in 300 µL of the working solution, and then incubated at 37 °C for 4 h. Three technical replicates of 100 µL were transferred to a 6-well plate and read at 492 nm absorbance using a Tecan infinite 200 pro spectrophotometer described previously [44,45].

### 2.10. Biofilm Viability Staining

Triplicate Ti samples were washed twice with phosphate-buffered saline (pH 7.2) at 24 and 72 h of culture before assessment of biofilm viability using a Filmtracer™ LIVE/DEAD™ Biofilm Viability Kit (Invitrogen, ThermoFisher Scientific, Scoresby, VIC, Australia) as previously described [46]. Following a 20 min incubation at room temperature (25 °C), the biofilms were washed once for removal of unbound stain and two-dimensional images of the biofilms captured using the confocal microscopy. Subsequently, 3D images were reconstructed with a step size of 2.0 µm.

### 2.11. Statistical Analysis

GraphPad Prism version 9.0.0 (Windows, GraphPad Software, San Diego, CA, USA) was used for all data analysis. All data are presented as mean and SD. The difference between the control, NaOH_Ti, and Li_Ti groups was analysed using two-way ANOVA with Tukey’s multiple comparisons test. The fold change for qPCR values was analysed using the 2^−∆∆Ct^ method. A *p*-value of <0.05 was considered statistically significant.

## 3. Results

### 3.1. Surface Characterisation of Ti Substrates

The surface topography of Ti substrates was characterised using SEM and the images are presented in Figure 1.

### 3.2. HGF Viability and Early Proliferation

Live/dead staining of HGFs over the sample groups showed no cytotoxicity signs after 1 and until 5 days of culture. DNA content was quantified after 4 and 24 h of culture to assess HGFs proliferation. More cells were present for both the Li_Ti and NaOH_Ti surfaces at 4 h compared to the control (untreated Ti) (Figure 2).

### 3.3. HGF Attachment and Morphology

Three-dimensional confocal microscopy images were used to view and analyse the HGF nuclei and actin filaments (Figure 3a–f) at 4 and 24 h post-seeding. Most cells were attached at 24 h in all groups, with no significant differences in nuclei count (Figure 4a). The measurements for the length and the aspect ratio (major axis of a cell/minor axis) were performed using ImageJ software (1.53f51, Wayne Rasband, Bethesda, MD, USA) (Figure 4b,c). Scanning electron microscopy images (Figure 3g–i) taken at 24 h confirmed an elongated and narrow cellular arrangement in the control group (spindle shape), compared to a wider, more branched appearance (stellate cells) of the HGFs in the treated Ti groups: Li_Ti and NaOH_Ti.

### 3.4. HGF Proliferation and Gene Expression of after 7 Days of Culture

After a longer incubation period (7 days), cells produced a denser and more irregular filament network on Li_Ti samples than other groups (Figure 5a). Both Li_Ti and NaOH_Ti surfaces induced significantly higher HGF proliferation than the control group (Figure 5b). Real-time PCR analysis (Figure 5c) demonstrated significantly increased expression of collagen I in both treated Ti groups compared to the control, and approximately an 8-fold increase in collagen I expression by HGFs on the Li_Ti surface compared to NaOH_Ti. Similarly, fibronectin was significantly increased in both treated Ti groups, with a six-fold increase in the Li_Ti compared to the NaOH_Ti group. The expression of collagen III, CXCL8 (interleukin 8) and IL1β (interleukin-1-beta) was higher in the Li_Ti and NaOH_Ti than in control, although not reaching statistical significance.

### 3.5. Analysis of Selected HGF-Secreted Proteins

Multiplex ELISA of conditioned culture media at 7 days (Figure 6) showed a significant increase in the concentration of FGF-2 in HGF cultures with Li_Ti and NaOH_Ti substrates, compared to the untreated control Ti (Figure 6a). Moreover, a significant decrease in MMP8 (Figure 6b) and VEGF (Figure 6e) was shown on days 3 and 7 compared to the control. No significant change was detected in MMP1 (Figure 6c) or PDGF-BB (Figure 6d). A summary of the *in vitro* assessments on HGFs bioactivity is illustrated in Table 2.

### 3.6. Analysis of Bacterial Metabolic Activity

For single-species biofilms, the metabolic activity of *S. aureus* in the Li_Ti group was the lowest after 1 and 3 days of culture and exhibited a significant difference to the alkaline group on the first-day post-culture. NaOH_Ti demonstrated slightly more bacterial activity than the control group on the first day. (Figure 7a). Metabolic activity of the salivary biofilms was significantly lower in Li_Ti than the control on days 1 and 3, and also showed remarkably fewer active bacteria than NaOH_Ti on day 3. (Figure 7c).

### 3.7. Biofilm Viability

Three-dimensional sections from stained samples were imaged under confocal microscopy to view live and dead single species and salivary biofilms over 1- and 3-days post-culture. Live/dead staining of single species (*S. aureus*) biofilms showed very few living bacterial cells in Li_Ti compared to the other groups on day 3 (Figure 7b). Similarly, salivary biofilms exhibited fewer viable bacteria for both Li_Ti and NaOH_Ti than the control surface, with the Li_Ti surface being the least favourable for viable bacteria, as shown by the red-stained areas 3 days post-culture compared to the control surface (Figure 7d).

## 4. Discussion

This study aimed to explore the effects of ECM-mimicking lithium-doped Ti nanostructure [39], on human gingival fibroblasts and oral biofilms. Previous studies have focused on the interaction between nanostructures of Ti surface and osteogenic cells in the context of osseous healing and osseointegration [9,10,11,12]. The current study focused on biocompatibility and anti-microbial properties of the Li-doped Ti surface from the peri-implant STI perspective.

### 4.1. Gingival Fibroblasts Response to the Surface

We hypothesized that the ECM-mimicking surface, doped with lithium ions [39], could positively influence the interaction of oral soft connective tissue cells. HGFs were chosen for the study, as they are the primary constituent cells in peri-implant connective tissue, responsible for forming the soft tissue seal against the oral environment [6]. HGFs produce adhesion proteins and ECM molecules essential in the soft tissue healing process, tissue attachment and formation at the transmucosal level [47].

Previous studies have shown that HGFs display enhanced proliferation on nanomodified substrates compared to micro-textured or smooth Ti groups [12,20,23]. The current results of increased viability actively formed an actin cytoskeleton, and the DNA production at a higher rate in HGFs cultured with nano-textured Ti substrates (NaOH_Ti and Li_Ti) (Figure 2 and Figure 3) corroborate these studies. The biocompatibility of the Li_Ti surface has been demonstrated previously by it promoting adhesion and growth of other cell types such as osteoblasts [39]. The nanowire-like mesh on the Ti surface with high resemblance to the collagen fibril arrangement in the ECM of native bone tissues considerably increased osteoblast viability, metabolism, adhesion, and proliferation. Isoshima et al. [48] used Li ions to create positive charges on the Ti surface for increased hydrophilicity, resulting in increased osteoblast attachment to the lithium charged surface, further supporting the potential benefit of these approaches for future clinical applications.

ECM formation is an important biological event for cellular attachment during the early phase of healing. After adhesion to the ECM surface, fibroblasts produce adhesion proteins such as collagen and fibronectin to ensure its structural support [49,50,51]. Collagen I is the main collagen type constituting the periodontal and peri-implant connective tissue structure [52]. In previous studies, Ti surfaces tuned with nanopores influenced the gene expression of collagen I [20,53]. Here, COL-I gene expression was shown to be significantly upregulated on the Li_Ti surface compared to both the NaOH_Ti and control-Ti surfaces (Figure 5c). A significant increase in the expression of fibronectin for the Li_Ti surface was also observed. Elevated fibronectin levels as an indicator of effective adhesion are well-established in the literature [54,55,56,57]. Indeed, it is the one glycoprotein produced by fibroblasts that regulates the adhesion process [58], acting as a “glue” for cell attachment. Together, these findings strongly support promotion of fibroblast metabolic activity by Li-induced surface nano-topography.

In our secretome analysis, the levels of VEGF and MMP-8 produced by HGFs cultured with both Li_Ti and NaOH_Ti substrates were significantly reduced compared to control (Figure 6). In previous tissue degradation models, VEGF inhibition was related to the reduction in collagen degradation [59], suggesting that the modified surfaces in the current study could potentially reduce collagenase activities [60]. Moreover, the level of FGF-2 secretion was significantly increased in the Li_Ti and NaOH_Ti cultures. FGF-2 is well known for its function in soft tissue healing and regeneration [61,62,63]. It is thus plausible that both the nano-scale topographies used in the present study could positively influence collagen production while reducing the expression of metalloproteinases.

### 4.2. Bacterial Activity over the Surface

The biofilm is considered the primary aetiological factor in the development and progression of peri-implant disease [4]. Complete eradication of pathogenic microbes in the oral environment is neither feasible nor realistic; however, considerable effort has been placed into developing surfaces that can restrain bacterial adhesion and growth, hence disturbing biofilm formation [64,65]. This is of clinical importance as the implant-transmucosal interface is where biofilm initially forms.

Our bacterial study was conducted using a single strain of bacteria (*S. aureus*) and multi-species bacteria collected from saliva. *S. aureus* is a commonly found bacterium on the skin and plays an essential role in the causation of medical device/implant-related biofilm infections [66,67]. The behaviour of mono-species biofilms in the lab is more predictable and controlled compared to polymicrobial biofilms. Hence, we initially chose to grow mono-species biofilms. However, most biofilm infections are polymicrobial, especially oral infections. So, the antimicrobial properties of the surfaces were evaluated against polymicrobial biofilms by using pooled saliva [68,69]. Quantitative data from our XTT experiment indicated that the bioinspired Li_Ti surface significantly reduced the early bacterial activity of both single and multi-species biofilm. Similarly, viability data for both microbial environments were consistent, showing a significant reduction in the bacterial growth in the Li_Ti group. Our results are in line with previous studies [17,18,33], in which nanoscale modifications on Ti substrates showed either bacteriostatic or bactericidal ability. Moghanian [70] reported that increased Li concentration in bioactive glass led to a prominent decrease in *Staphylococcus aureus* activity. This was compatible with our bacterial activity study, where the Li-containing substrate exhibited increased suppression of *S. aureus* activity compared to the alkaline group (Figure 7a). Moreover, the metabolic activity of the multi-species salivary biofilm was significantly reduced on the Li_Ti surface compared to NaOH_Ti (Figure 7c). Importantly, the present study is the first to investigate the antibacterial effect of Li on a multi-species biofilm model.

In addition to HGFs, peri-implant soft tissue is composed of other cell types including epithelial cells and innate immune cells, and hence the current work’s sole focus on HGF response to the modified Ti substrates may be considered a study limitation. It would be of importance to investigate all cell responses to the Li_Ti modified surface from a clinical perspective. Our bacterial culture study being conducted under a static condition is a second limitation. The flow of saliva in the oral cavity, as simulated in a dynamic model, may influence bacterial activity and survival on Ti surfaces not accounted for in our current static model. Nevertheless, the current work is the first to demonstrate the antibacterial characteristics of the nano-modified Ti surface by using multi-species biofilm, rather than single species bacteria alone.

Biocompatibility and antibacterial effects of the nano-modified Li_Ti surface should be further investigated in an in vivo environment, preferably in an oral environment, to provide a better understanding of the biological and microbiological mechanisms of the surface, therefore allowing the exploitation of its potential for clinical application.

## 5. Conclusions

A dental implant surface capable of augmenting the function of gingival fibroblasts and reducing the adhesion of bacteria may enable the early establishment of STI and increase long-term survival. Remarkably, the Li-doped Ti (Li_Ti) surface resulted in upregulated expression of COL-I and fibronectin compared to the Ti nanostructure without lithium (NaOH_Ti). In addition, the Li_Ti surface promoted an increase in the concentration of growth factors (FGF2), while significantly reducing collagenase (MMP8) and VEGF secretion compared to the control Ti surface. Concerning its effects on bioactivity, the bioinspired Li_Ti surface can augment HGF cellular attachment, proliferation, collagen formation, and extracellular matrix deposition. As for antibacterial activity, both treated Ti (nanoscale modified topographies) surfaces significantly reduced bacterial adhesion and growth compared to the untreated smooth machine polished (control) Ti surface. These antibacterial effects were more evident at day 3 for the Li_Ti surface compared to the control group. As such, it may be concluded that the bioinspired Li-doped Ti surface can promote HGF bioactivity while suppressing bacterial adhesion and growth. This is of clinical importance in terms of improved STI during the maintenance phase of implant treatment. Further *in vivo* studies are warranted to investigate Li-doped surfaces’ effects on the host immune responses and tissue formation quality.

## Figures and Tables

**Figure 1 nanomaterials-11-02799-f001:**
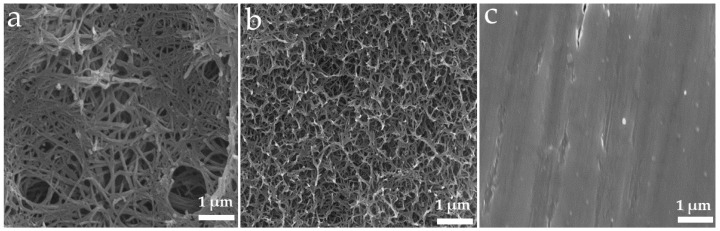
Top view SEM images showing the surface of Ti substrates. (**a**) Lithium-incorporated Ti (Li_Ti), (**b**) alkaline-treated Ti (without Li) (NaOH_Ti) and (**c**) mechanically micro-machined Ti (Control_Ti). All scale bars represent 1 um.

**Figure 2 nanomaterials-11-02799-f002:**
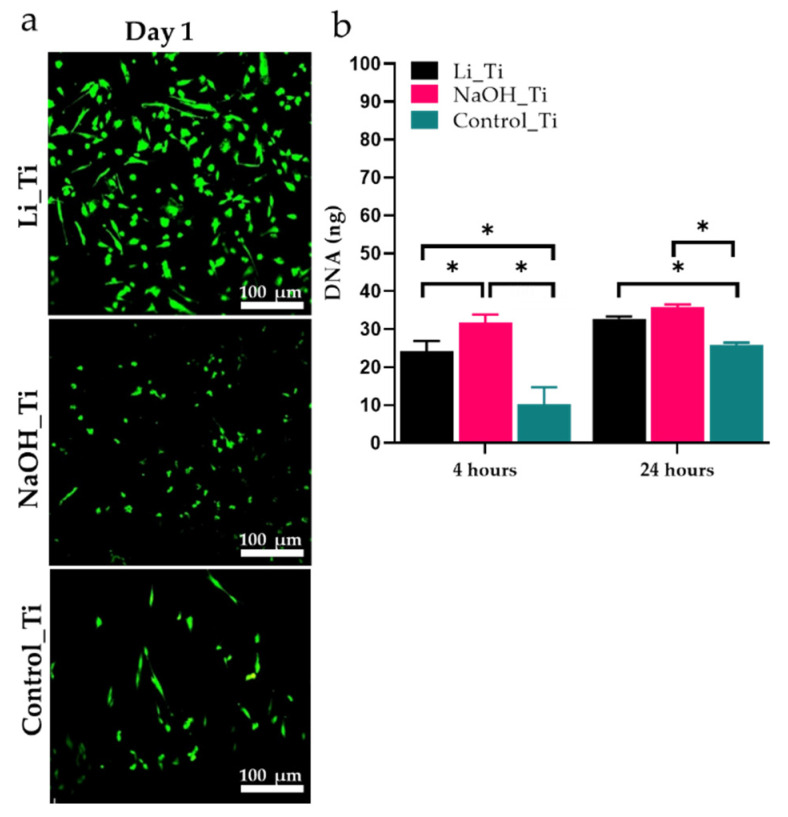
Viability and proliferation of human gingival fibroblasts. (**a**) Confocal microscopy images of Live-Dead staining over Li_Ti, NaOH_Ti and Control_Ti substrates at day 1, (**b**) analysis of PicoGreen assay for DNA content. * *p* < 0.05.

**Figure 3 nanomaterials-11-02799-f003:**
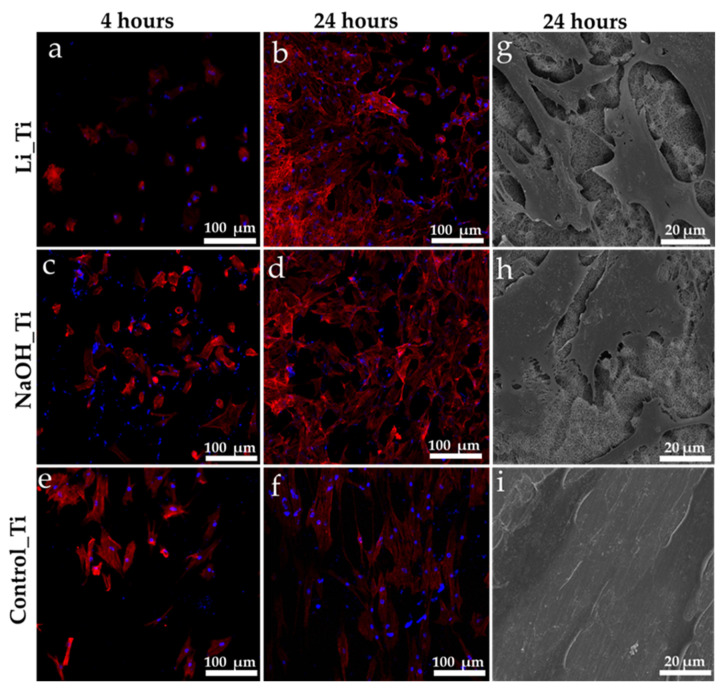
Human gingival fibroblasts’ attachment. (**a**–**f**) Nuclei (blue) and actin F (red) staining using confocal microscopy images of HGFs at 4 and 24 h, (**g**–**i**) SEM images of the HGF over Ti groups, ×1000 magnification.

**Figure 4 nanomaterials-11-02799-f004:**
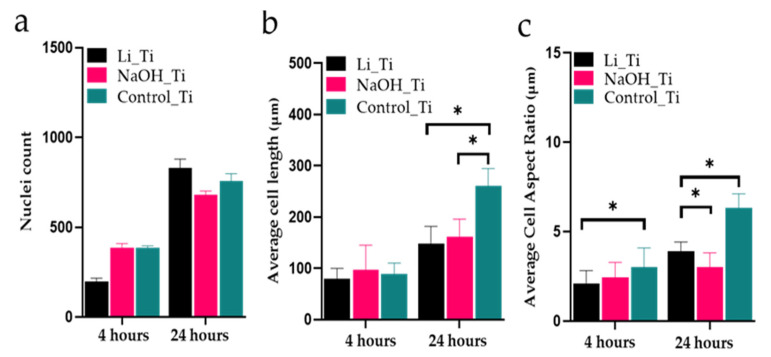
Cell morphology analysis at 4 and 24 h culture. (**a**) Nuclei counts, (**b**) cell length, and (**c**) length-to-width ratio (Aspect Ratio). Three-dimensional confocal microscopy images in Figure 3 were used for the analysis, * *p* < 0.05.

**Figure 5 nanomaterials-11-02799-f005:**
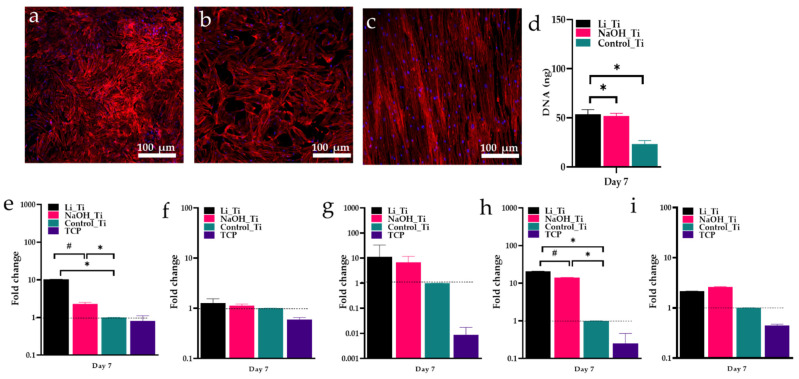
Human gingival fibroblasts characterisation 7 days after culture. (**a**–**c**) Confocal microscopy images of HGF at day 7 showing nuclei (blue) and Actin F (red) staining, (**d**) PicoGreen assay for DNA content, (**e**–**i**) HGF gene expression showing the fold change of COL-I, COL-III, CXCL8, fibronectin and IL1β. Dotted lines refers to the reference value * *p* < 0.05; # *p* < 0.01.

**Figure 6 nanomaterials-11-02799-f006:**
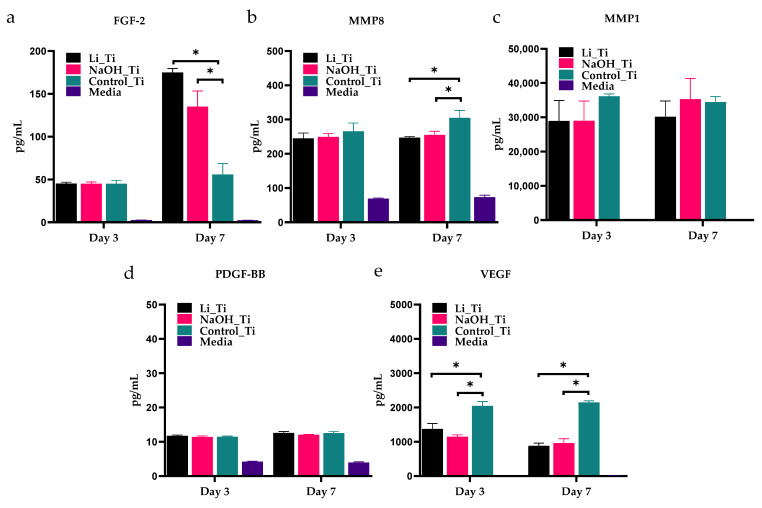
Multiplex ELISA quantification of conditioned media concentrations of FGF-2 (**a**), MMP-8 (**b**), MMP-1 (**c**), PDGF-BB (**d**) and VEGF (**e**) from HGF cultures with Ti substrates. * *p* < 0.05.

**Figure 7 nanomaterials-11-02799-f007:**
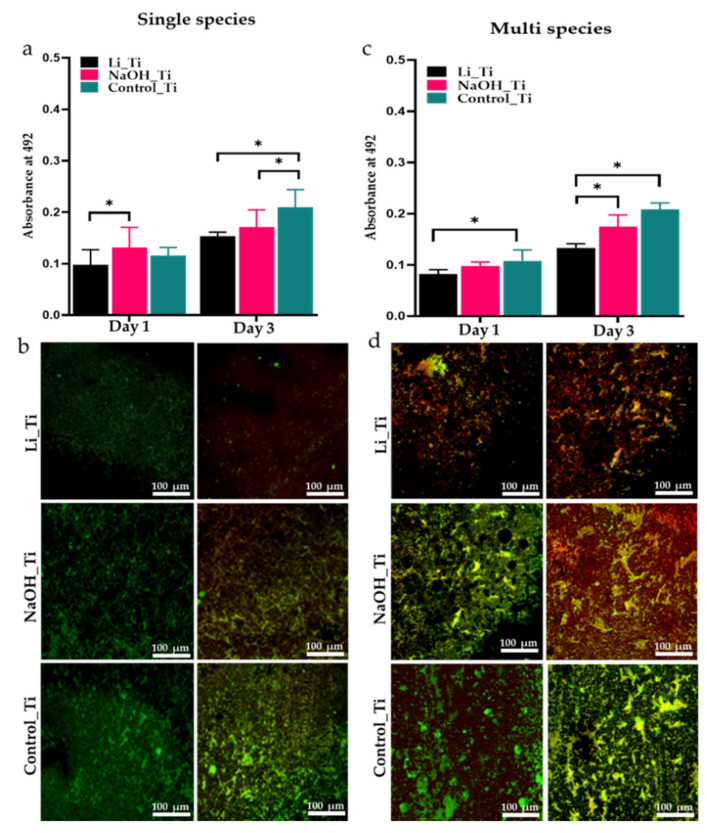
Metabolic activity (**a**,**c**), and live (green)/dead (red) staining (**b**,**d**) of single species (*S. aureus*) and multispecies (saliva) biofilm on three Ti substrates (Li_Ti, NaOH_Ti and Control_Ti) after 1 and 3 days of culture. * *p* < 0.05 by two-way ANOVA with Tukey’s multiple comparison test.

**Table 1 nanomaterials-11-02799-t001:** The experimented genes’ symbols and primer sequences in forward 5′-3′ and reverse 3′-5′, and length in base pair (bp).

Gene Symbol	Direction	Primer Sequence	Length (bp)
hCOL1A1	Forward	CCTGCGTGTACCCCACTCA	115
	Reverse	ACCAGACATGCCTCTTGTCCTT	115
hCOL3A1	Fwd	CCGTTCTCTGCGATGACATAA	142
	Rev	CCTTGAGGTCCTTGACCATTAG	142
hGAPDH	Fwd	TCAGCAATGCATCCTGCAC	117
	Rev	TCTGGGTGGCAGTGATGGC	117
h18S	Fwd	CAGACATTGACCTCACCAAGAG	99
	Rev	GAATCTTCTTCAGTCGCTCCAG	99
hIL_1B	Fwd	GGTGTTCTCCATGTCCTTTGTA	125
	Rev	GCTGTAGAGTGGGCTTATCATC	125
hCXCL8	Fwd	GAGAGTGATTGAGAGTGGACCAC	112
	Rev	CACAACCCTCTGCACCCAGTTT	112
hFN1	Fwd	CACAGTCAGTGTGGTTGCCT	68
	Rev	CTGTGGACTGGGTTCCAATCA	68

**Table 2 nanomaterials-11-02799-t002:** Summary of the bioactivity assessments of varied Ti implants.

Figure	Test/Assay	Time Points	Description	Inference
Figure 2a	Livedead staining	D1	Viability of cells	Live cells were observed in all groups ( no signs of cytotoxicity)
Figure 2b	Picogreen	D1	Early cell proliferation measured by DNA content	Some significance in DNA content was observed
Figure 3a–f	Immunofluorescence staining (DAPI, phalloidin)	4 h, D1	Early visualization of nuclei and actin filaments	Generated images (at least 3 samples per group) were used for the analysis showed in Figure 4
Figure 3g–i	Scanning electron microscopy	D1	Detailed information of the surface and attached cells	Closer visualization of cellular morphology
Figure 5a–c	Immunofluorescence staining (DAPI, Phalloidin)	D7	1 week old visualization of nuclei and actin filaments	Some difference of the filaments density was observed
Figure 5d	Picogreen	D7	1 week old cell proliferation measured by DNA content	Significance of DNA content between groups
Figure 5e–i	Real time PCR	D7	Quantification of mRNA levels of selected primers	Significant increase in the expression of COL 1 and Fibronectin between Li_Ti and NaOH_Ti
Figure 6	Multiplex-ELISA	D7	Quantification of protein concentrations in the culture media	Significant difference in protein concentration between treated titanium groups vs. control titanium

## Data Availability

The data are not publicly available.

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
