# Peer review of "Influence of Bioinspired Lithium-Doped Titanium Implants on Gingival Fibroblast Bioactivity and Biofilm Adhesion"

_nanomaterials, 2021, doi:10.3390/nano11112799_

Round 1
Reviewer 1 Report
SEE REPORT

Author Response
Thanks for the reviewer's comments. Kindly find attached our responses to the comments.

Reviewer 2 Report
- The mechanisms of the biochemical effects of lithium are diverse and include the effect on the metabolism of sodium, calcium, magnesium, changes in the functions of various enzymes, hormones, vitamins and growth factors. Lithium in certain amounts can be harmful to health.
- Line "Nanostructured Ti" is usually used in the case of titanium with a grain / subgrain size of about 100 nm, and has ultra-high strength. In your case, it was coarse-grained titanium.
- Line 236. SEM images (Figure 1) - there are no numbers on the measured segments. Usually, the numbers are indicated in the pictures.
- Line 239. In the literature, it is often mentioned that wells of about 1 μm are favorable. In your results in Figure 1a and 1b, the size of the structure elements is different, but the DNA content is very similar (Figure 2). The reason?
Author Response
Thanks for the reviewer's feedback. Kindly view our response in the attached word file. A newer version of the manuscript has been uploaded as well.

Reviewer 3 Report
General comments
It is an interesting paper describing a good research with a potential for useful applications
It seems almost ready for publication, but there are some misprints and small suggestions.
Particular comments.
(i) Lines 17 and 204 – it seems that additional square bracket ‘[‘ erroneously appeared in the text
(ii) Abstract, Keywords. Consider adding ‘surface properties’
(iii) line 60: “…Moreover, these features influences initial filopodia-surface interactions…” conflict of plural an singular. Suggested: “…Moreover, these features influence initial filopodia-surface interactions…”
(iv) line 89. “… (equal volumes of concentrated H2SO4 : HCl: H2O)..” is a little confusing. Suggestion: “… (equal volumes of concentrated acids and water H2SO4 : HCl: H2O)..”
(v) line 111: “…at a density of 5000 per Ti substrate…” suggested “…at a density of 5000 cells per Ti substrate…”
(vi) line 134: “…in two changes of ethanol (20-100%)…” some confusing. If I understand correctly, it would be better to use: : “…in two changes of ethanol (20% and 100%)…”
(vii) line 159, Table 1, ‘Length’ column- please, state the units for the length
(viii) lines 236, 237- Capture to Figure 1. Two corrections suggested, resulting in: “…Top view SEM images showing the surface of Ti substrates. a) Lithium incorporated Ti (Li_Ti), b) Alkaline treated Ti (without Li) (NaOH_Ti) and c)…”
(ix) lines 270 and 272, description of Figure 3. It seems that references ‘Figure 4s’ and ‘Figure 4b, c’ are incorrect!
(x) line m451, suggested to remove ‘fold’ from the sentence, to have it like “…A significant increase in the expression…”
(xi) lines 475, 476- sequence of tenses, should be ‘reduced’ instead of ’reduce’?
Scan of the paper with all handwritten remarks will be available from the Editorial system

Author Response
Thanks for the reviewer's comments. They have all been addressed in the newer version of the manuscript.
As for the comment"(ix) lines 270 and 272, description of Figure 3. It seems that references ‘Figure 4s’ and ‘Figure 4b, c’ are incorrect!"
Figure 3 (1-i) confocal images were used to produce the analysis presented in Figure4 (a,b,c). Thus, we referred to both of the figures in the same paragraph, as they are related.

Round 2
Reviewer 1 Report
see attached file

Author Response
Thanks for the reviewer's comments. Kindly find attached the revised manuscript.
